# Physical Fitness, Executive Functions, and Academic Performance in Children and Youth: A Cross-Sectional Study

**DOI:** 10.3390/bs14111022

**Published:** 2024-11-01

**Authors:** Valter Fernandes, Arthur Silva, Andrea Carvalho, Sidarta Ribeiro, Andrea Deslandes

**Affiliations:** 1Exercise Neuroscience Laboratory, Institute of Psychiatry, Federal University of Rio de Janeiro, Rio de Janeiro 22290-140, RJ, Brazil; 2Instituto Nacional de Tecnologia, Rio de Janeiro 20081-312, RJ, Brazil; 3Brain Institute, Federal University of Rio Grande do Norte, UFRN, Natal 59076-550, RN, Brazil; 4Mental Health Research Group, Center for Strategic Studies (CEE), Fiocruz 21040-361, RJ, Brazil

**Keywords:** sport skills, motor development, academic success, cognition

## Abstract

The aim of this cross-sectional study was to investigate the relationship between physical fitness, executive function, and academic performance in children and adolescents. A total of 131 students (49% female) aged 10–15 years from a public school in Rio de Janeiro were assessed in executive functions (hearts and flowers, Corsi’s block, and digit span tasks), academic performance (Portuguese, reading, math, and overall school grade), physical tests (touch test disc, agility, lower limb and upper limb explosive strength), and anthropometric measurements. Regression results showed that the composite of sports-related fitness measures was the best predictor of executive functions (β = 0.472; t = −6.075 *p* < 0.001). Decision tree classifier analysis showed that the combination of factors that discriminated better and worse executive function groups were better performance in hand–eye coordination (TTD), math, and upper limb strength (ULEST). Sports-related fitness is significantly correlated with executive function. Hand–eye motor coordination has been identified as the most important predictor of improved cognitive outcomes, surpassing even academic skills. These findings should be considered in the design of physical activity programs in school settings, which may have a positive impact on child development, reflected in the reduction of academic and socioeconomic disparities.

## 1. Introduction

High levels of physical activity are directly related to lower maximum body mass index (BMI) and greater cardiorespiratory capacity. Both are associated with various health benefits and improved cognitive performance across the lifespan. Regular sports practice goes a step further by adding the development of physical skills such as agility, speed, and motor coordination, which are essential not only on the courts and fields but in many activities of daily living [1]. There was a comprehensive study conducted in the United States that examined youth risk-taking behavior. The findings revealed a strong association between participation in team sports and improved academic achievement [2].

On the other hand, sedentary children performed worse on tests of inhibitory control than physically active children [3]. Inhibitory control is part of a set of cognitive skills called executive functions, which further comprises working memory, cognitive flexibility, and high-level skills such as planning and decision making [4]. Such skills are essential for almost all aspects of human life, such as social and affective relationships, learning, and academic performance [5].

Childhood is a crucial period for cognitive development, as stimuli at this stage will have lifelong effects [6]. Among the multiple cultural aspects that leverage childhood development, exercise has been highlighted by the literature for producing significant benefits for executive functions [7]. However, the literature still lacks stronger evidence and a better understanding of how these outcomes may be mediated and moderated by different variables of exercise, the context of the intervention, and the characteristics of the children. In addition, evidence is also lacking to clarify which aspects of fitness need to be further encouraged so that these benefits can extend beyond the courts and fields.

To elucidate these gaps, researchers have been producing cross-sectional studies to investigate the correlations between physical and cognitive abilities. The first studies, and still the majority in this field, focused their efforts mainly on the relationship between cardiorespiratory capacity and brain structure and function [8]. A recent meta-analysis demonstrated a positive correlation between aerobic fitness and academic performance in obese and overweight children, but this correlation was stronger in boys than in girls [9]. In addition to health-related aspects of physical fitness, motor competence appears to be intrinsically related to full and healthy cognitive development. Studies have shown that children that are more agile tended to perform better on tests of inhibitory control and cognitive flexibility [10]. More complex motor tasks, usually with timed movement performance, bring greater attentional engagement, requiring greater cognitive effort [11]. In a study with Brazilian children, our research group showed that children with better scores on a timed hand–eye coordination test tended to have better scores on tests of academic achievement and selective attention [12].

Despite the relevance of the relationship of more complex physical skills with cognition, the literature does not have a consensus on which skills are more relevant for different aspects of children’s cognition. One study with kindergarten children highlighted the relationship between behavioral self-regulation and executive functions with fine motor skills. However, they were differentially related to different aspects of academic performance [13]. In a study of obese and overweight children, muscular strength, agility, and cardiorespiratory fitness were associated with tests of executive function, but the cognitive flexibility test showed the most significant association with all fitness components [14]. Another study, with children aged 9 to 13 years, found significant associations between physical fitness and executive functions. In addition, indirect associations were observed between physical fitness and academic performance, mediated by executive functions [15].

Thus, this study aims to investigate the correlation between physical fitness, executive functions, and academic performance in children and adolescents. We hypothesize that superior performance in assessments of sport-related physical skills, such as hand–eye coordination and agility, will exhibit stronger correlations with improved performance in assessments of executive function and academic skills among children and adolescents. Understanding how different physical and cognitive abilities relate to each other may help in designing school physical education programs with pedagogical objectives capable of extending the results from the courts to the classrooms.

## 2. Methods

### 2.1. Participants

The sample consisted of 131 students from Noel Nutels Municipal School, 10 to 15 years old, 64 (49%) females and 67 (51%) males. No scientific studies were found in the academic literature that shared all the outcomes of this study with a sample of Brazilian children, which made it impossible to calculate the sample size. Thus, the number of participants estimated in this study was determined by the structural conditions of the participating schools, being characterized as a convenience sample.

The participants and their guardians were informed about the experimental procedure, signed the Informed Consent Form and the Terms of Consent, and underwent an initial anamnesis. Data from children who had uncorrected hearing and visual impairment were not included in this study. The assessments were conducted from February to the first week of March 2020. This project was approved by the Ethics and Research Committee of the Centro Universitário Augusto Motta (UNISUAM) (CAAE 11980119.4.0000.5235).

### 2.2. Measures

#### 2.2.1. Executive Functions

Computerized cognitive assessments were performed on Inquisit 5 Lab software from the millisecond.com platform (https://www.millisecond.com (accessed on 12 September 2024)). The executive function test battery described below was performed on iPads, lasting a maximum of 30 min.

Participants completed the hearts and flowers task (HFT), composed of congruent and incongruent commands, characterized by assessing several core aspects of executive functions, and in particular, cognitive flexibility. When a heart appears on the screen, one must tap on the same side, and when a flower appears, one must tap on the opposite side of the screen as the figure appears on the screen. The hearts and flowers task includes the following phases: blocks with hearts only, blocks with flowers only, and blocks with a mixture of hearts and flowers. In this study, we used the mixed blocks phase, recording reaction time and accuracy to calculate the time cost score for correct answers (HFT), with the following calculation: (average reaction time/percentage of correct answers)/100 [16].

In the digit span task (forward and backward), the direct order task tests short-term memory and serves as an introductory task for the reverse order task, considered a task for working memory, one of the central executive functions. In both steps, participants hear the digit sequences and need to retrieve them (in a forward or backward order) by selecting digits from a circle of digits by touching the touchscreen. Depending on their performance, the participants were either moved up or down one level. The evaluation ends after 14 trials. We used the sum of the measures of the maximum length up to two errors (TE_ML) from the forward and backward phase responses as a measure of working memory [17].

Corsi’s block assesses visuospatial short-term memory and is performed by presenting images of cubes, which appear on the screen in a predetermined sequence (constant across participants), and participants must click on the cubes in the same order as they were presented. The sequence duration starts at level-two cubes and can increase up to level-nine cubes. Participants have two chances in each sequence duration. If one of the sequences is answered correctly, the next sequence is started. We use the total score measure that corresponds to the multiplication of the number of items correctly recalled during the entire task by the length of the last correctly answered cube sequence [18].

#### 2.2.2. Academic Performance

The reading subtest of the School Performance Test (SPT-R) was used as a measure of academic performance. The task consists of the sequential reading of 70 words isolated from context [19]. In addition, math, Portuguese, and the overall school grades (average of all subjects) were collected from the previous school bimester. Thus, a composite score of academic performance (AcadC) was generated by combining the school grades score with the SPT-R.

#### 2.2.3. Physical Assessments

The following anthropometric measures were collected as part of the physical assessment: body mass, height, trunk height, and waist circumference. From the collected data, we estimated the body mass index (BMI) and predicted years from peak height velocity, a maturity offset value [20]. In addition to these, tests of aspects of physical fitness, directly related to sports performance of children and adolescents, were performed.

Upper limbs explosive strength test (ULEST)—The student sits with knees extended, legs together, and back fully supported against a wall, and holds a 2 kg medicine ball close to their chest with their elbows bent. At the signal from the evaluator, the student must throw the ball as far as possible while keeping their back supported against the wall. The distance of the throw is measured from the wall to the place where the ball first touches the ground. Two throws are made, and the best result is recorded in centimeters [21].

Lower limb explosive strength test (LLEST; horizontal jump)—The student is positioned behind a starting line, which can be marked with tape on the floor or one of the lines that mark the sports courts. At the signal, the student must jump as far as possible, landing with both feet at the same time. Two attempts are made, and the best result is recorded in centimeters [21].

Agility test (square test)—A square of 4 by 4 m is marked out on the test site with lines marked with tape on the ground and a cone in each corner. The student stands in one of the cones and waits for the evaluator’s signal to run as fast as possible, touching each cone with one hand, crossing two diagonals and two straight lines until all cones are touched. Two attempts are made, and the shortest time is recorded in seconds and hundredths of a second [21].

Touch test disc (TTD)—Evaluates motor coordination, particularly hand–eye coordination. The test is performed on a rectangular wooden board 120 cm by 60 cm wide. In the center of the board is a rectangle 10 cm high by 20 cm wide and a circle 20 cm in diameter on each side, with 5 cm between the figures. The subject must keep the non-dominant hand in the central rectangle and touch the circle on the opposite side with the dominant hand, crossing one arm over the other, and return to complete a cycle. Each attempt consists of 25 correct cycles, and the minimum time for completion of three attempts is counted [12].

### 2.3. Procuderes

The evaluations were performed on three different days: day 1—registration, anthropometric evaluations, and the SPT-R; day 2—physical evaluations; day 3—computerized cognitive evaluations. The evaluations were extended for one more day whenever the evaluators noticed fatigue or discomfort on the part of the participants. The team of assessors was made up of psychologists and physical education teachers, who were properly trained in the instruments of this research.

### 2.4. Statistical Analysis

The Kolmogorov–Smirnov test was used to assess the normality of the variables. The Mann–Whitney test was used for descriptive analysis of the dependent variables shown in Table 1, with mean and standard deviation values for each study variable. The composite score variables of physical fitness (FitC—the sum of the inverted percentiles of TTD and agility, as well as the percentiles of LLEST and ULEST, so all went in the same direction) and academic performance (AcadC—the average of school grades score + SPT-R ((SPT-R/70) × 10)) were created.

To verify the relationships between the motor and cognitive variables, Pearson and Spearman correlations were performed. Then, tests of correlations paired by the sex and age of the subjects were performed. First, the database with the imputed data was used, and then we performed a paired correlation without the imputed data. A Bonferroni correction adjusted the *p*-value against the number of correlations that were performed. Linear regression was used to analyze the unstandardized coefficient of variables capable of predicting performance in executive functions. The above-mentioned statistical analyses were performed using SPSS for Windows, Version 26.

To enhance comprehension of the measured correlations, we employed machine learning (ML) techniques to create a model capable of identifying a set of features that indicate a higher likelihood of improved cognitive performance. Among the techniques considered, the decision tree algorithm was chosen for its interpretability, so we can better examine and understand the relationship between the chosen input variables and the outcomes of interest. This algorithm allows for a clear and understandable representation of the decision-making process, making it easier to interpret and analyze the results [22].

To optimize the sample, the data were first imputed using the K-nearest neighbors (KNN) technique [23]. This technique helps fill in missing data points by considering neighboring data points. Subjects were categorized into better (≥50%) or worse (≤50%) performance in executive functions according to percentile on the HFT. After that, the values of all data were standardized to a range between zero and one, and then decision tree classification models were trained on these imputed and standardized data (imputation and standardization were not performed on the test data to avoid data leakage). We set parameters to prune the trees, ensuring that their maximum depths were limited to three. Divisions and leaf nodes were only created if the number of occurrences exceeded ten, ensuring the sample remained representative.

During our analysis, we performed several tests using decision trees to identify the most effective predictors. The outcome variable we focused on was a binary variable that classified individuals as having high (high EF) or low (low EF) performance in executive functions, considering HFT. The models generated in this study were evaluated based on multiple metrics, including accuracy, precision, and recall. The aim was to select a model that demonstrated a balanced performance across these metrics. The software used to build this model included Python version 3.11.1 and the packages “sklearn” (http://scikit-learn.org/stable/index.html (accessed on 12 September 2024)) and “pandas” (https://pandas.pydata.org/ (accessed on 12 September 2024)).

## 3. Results

### 3.1. Sample and Data Description

As for the variables of physical fitness for sport, data inspection verified only 2 (1.53%) missing pieces of data in the TTD and 75 (57.25%) in the grouped data of the other variables of the physical fitness composite score. As for the cognitive measures, 10 (7.63%) missing pieces of data were found in the HFT, 40 (30.53%) in the Corsi task, and 4 (3.05%) in the digit task. On the Portuguese and math scores, 63 (48.09%) subjects had missing data, while on the academic performance composite score, 55 (41.98%) subjects had missing data. The Kolmogorov–Smirnov test showed that only the TTD, FitC, and math scores had parametric data. Table 1 presents the total sample data of the 131 subjects, including the imputed data, categorized and compared between male and female participants. Regarding PHV, the results showed that girls had significantly lower PHV ages than boys, demonstrating more accelerated maturation. As for the physical tests, the boys showed significantly better results only in the agility, LLEST, and the FitC tests. The cognitive and academic performance tests showed no significance after the Bonferroni correction (see Table 1).

### 3.2. Correlations

Spearman’s correlation showed significant results between the HFT and the agility measures (0.420, *p* < 0.001), TTD (0.336, *p* < 0.001), and FitC (−0.457, *p* < 0.001) (see Table 2). Surprisingly, the Portuguese scores had negative and significant correlations with ULEST (−0.301, *p* < 0.001) and FitC (−0.457, *p* < 0.001). Likewise, the correlation between subjects’ age and math (−0.294, <0.001) scores also showed significantly negative results.

Table 3 shows the results of the correlation tests controlled for sex and age from the original data sample and the database with imputed data. The results were significant between the HFT and the hand–eye coordination measure TTD, maintaining the significance level (*p* < 0.001) for both the imputed data and the original data. Correlations between the HFT, agility test, and FitC, on the other hand, were significant with the imputed data, but with the original database they were just above the significance threshold (*p* = 0.004). The correlations between the Portuguese scores and the FitC were not significant, but with the imputed database, the result was also close to the significance threshold (*p* = 0.004). The comparison between the controlled correlations of both databases showed results in the same direction, pointing to similar correlations in both strength and direction (Table 3). Linear regression analysis resulted in a statistically significant model (F(1,129) = 36.90; *p* < 0.001; R^2^ = 0.22) and corroborated the regression results, showing FitC as a predictor of executive functions (β = 00.472; t = −6.075 *p* < 0.001).

### 3.3. Machine Learning Analysis

In order to use ML techniques in the search for an algorithm that could better predict cognitive performance, the subjects were categorized into high (high EF) and low (low EF) performance in executive functions, considering the HFT performance (see Appendix A). In addition to the differences in HFT, after Bonferroni correction, there were differences (*p* < 0.001) in the agility test, TTD, FitC, and AcadC, with significantly better results for the high-EF group.

After exploring several ML techniques, the model that led to the best accuracy was a decision tree classifier (DTC) model, with the categorical HFT variable as the response variable [24]. Figure 1 presents the results of the chosen decision tree model, showing higher sensitivity for students with high EF than students with low EF. The trained model achieved an accuracy of 71%, precision of 70%, and recall of 82%. The classification scores revealed that the best features for distinguishing between the high- and low-executive-function (EF) groups are the combined performances in the TTD, math, and ULEST variables.

## 4. Discussion

The aim of this study was to investigate the correlation between physical fitness, executive functions, and academic performance in children and adolescents. The results of the regression analyses showed that FitC, the composite of sports-related fitness measures, was the best predictor of executive function. The DTC analysis showed that the combination of features that discriminated the better- and worse-performing groups in executive function were the performances in hand–eye motor coordination (TTD), math performance, and upper limb strength (ULEST). The results highlight hand–eye coordination, one of the components of FitC and a skill of great importance for sports practice with a strong correlation with childhood cognition [12]. Based on the DTC analysis, which includes measures of academic achievement, it appears that hand–eye coordination is the most significant component among the predictors associated with better executive function performance. To the best of our knowledge, this is the first study to attempt to verify such a hypothesis through a decision tree classification method using ML algorithms.

Initial correlations showed significant results between executive functions and measures of agility, hand–eye coordination, and composite fitness score. These findings provide support for our hypotheses and corroborate the evidence indicating that motor skills, particularly those requiring precise task execution, exhibit stronger correlations with more complex cognitive abilities such as executive function [25]. Despite this, surprisingly, Portuguese scores had negative and significant correlations with ULEST and FitC. Similarly, the correlation between age and math scores indicated that, with increasing age, scores tended to decrease. Indeed, there are several nuances in the relationship between different aspects of physical fitness and cognition, which change throughout development and the maturational process. A longitudinal study with Norwegian children showed that at age 9 years there was a low correlation between higher motor competence and reading skills in the whole sample, but specifically for boys, this correlation was inverse [26]. In our study, boys scored significantly better on the physical fitness variables tested, while girls scored better on Portuguese scores, although not significantly after Bonferroni correction. Differences in levels of participation in physical activity [27], as well as different rates of maturation between boys and girls [20], may contribute to our understanding of these findings.

The correlations controlled for sex and age corroborate our hypotheses and help us in elucidating the differences in performance between sexes and different maturational levels. The results were significant for the correlations between executive functions, measured by the HFT, and the TTD hand–eye coordination measure. The controlled correlations maintained the level of significance (*p* < 0.001) for both the imputed and original data. However, the correlations between the HFT, the agility test (*p* = 0.004), and the FitC (0.004) were just above the significance level after Bonferroni correction in the original data but reached significance in the imputed data. The TTD had only 1.53% of the imputed data, while the agility test and the other FitC measures had 57.25% imputed. We can understand these results as a positive verification of the data imputation method, since the strengths and directions are congruent and the differences are pointed out by the differences in the N of calculating the correlations.

Our results are in agreement with studies that describe how neural networks related to hand–eye coordination interact with the function of attentional control, involving the prefrontal cortex, cerebellum, and basal ganglia [28,29]. A physical exercise program that requires open skill (performing complex motor tasks in an unpredictable environment that requires focused attention while receiving multiple sensory stimuli) can provide significant improvements in hand–eye coordination and cognitive skills [30,31]. A recent study showed that in addition to motor coordination being strongly associated with accuracy in visuospatial working memory tasks, increased motor ability was related to higher P300 evoked potential amplitude in the centro-parietal cortex [32]. Neural networks underlying visuospatial working memory, sensory-motor adaptation, and the learning of motor sequences are supported by increased activation of the frontoparietal network—i.e., the right dorsolateral prefrontal cortex and bilateral posterior parietal cortices [33].

This circuitry seems to be crucial in the retrieval process of already assimilated motor gestures, which will be coupled to new gestures that will compose, for example, a more complex sport skill (e.g., a side kick with the sole of the foot, added to the rotation of the trunk, for the spinning kick). Similarly, the systems reliant on the visuospatial draft and the phonological loop, which are responsible for temporarily storing visual, spatial, phonological, and auditory information, respectively, will be triggered. Not coincidentally, interventions that aimed to stimulate hand–eye coordination and visuospatial attention showed significant results in motor coordination [31], executive functions [29], and academic performance [34].

Comparing the results between the sexes, the maturational age analysis showed a more accelerated maturation of girls. Despite this, boys showed significantly better results in the agility, LLEST, and the FitC tests, but not in the cognitive and academic tests. In fact, girls tend to reach peak high velocity at age 11, while fewer do so at age 13 [6]. Girls’ worse performances on fitness tests may be related to lower rates of physical activity and low participation in exercise programs [35].

## 5. Conclusions

Our study brings important evidence, which may influence the teaching practice of physical education professionals, extending its potential effects to other school subjects. However, some limitations of the study deserve to be highlighted. The great amount of missing data, imposed by the reality of conducting a study in a school environment, weeks before the quarantine imposed by the COVID-19 pandemic, is undoubtedly an important limitation of the study. In addition, the lack of assessments of physical activity levels, as well as type of sports participation and socioeconomic status, also limits the understanding of these possible moderators of outcomes. However, the use of the data imputation technique allowed us to reach significant results, corroborated by the comparative results of the correlations with the original data.

Sport-related fitness is significantly correlated with executive function. Hand–eye motor coordination is the most important predictor of better cognitive results, even in comparison with academic skills. These findings should be considered when designing physical exercise programs in the school environment, which may ensure positive impacts on child development, reflecting the reduction of academic and socioeconomic inequalities. Experimental studies should be designed to stimulate sports-related fitness skills, in order to investigate potential effects on children’s cognition.

## 6. Practical Implications

Developing sports-related fitness is a potential strategy to reduce the gap in both cognitive development and academic achievement.Hand–eye motor coordination is the best predictor of better executive function, even in comparison to academic skills.Physical education programs should consider the importance of improving hand–eye coordination to extend its benefits to cognitive development and academic performance.

## Figures and Tables

**Figure 1 behavsci-14-01022-f001:**
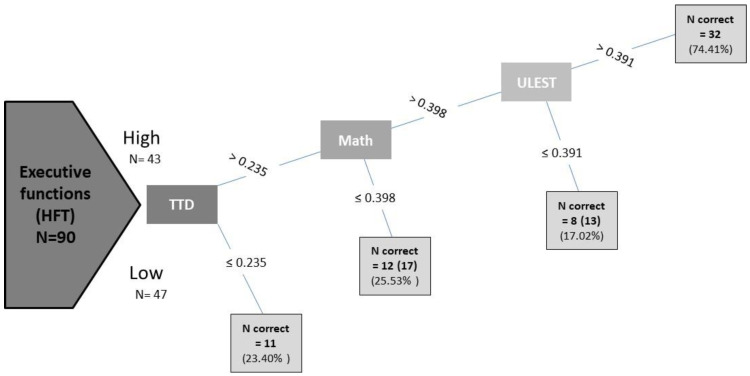
Decision tree classifier. HFT (inverted cost of correct responses for the hearts and flowers task—executive functions); high and low performance categories in HFT; TTD (touch test disc); ULEST (upper limb explosive strength test); N correct and % N categorical (high = 43; low = 47).

**Table 1 behavsci-14-01022-t001:** Descriptive analysis of the sample categorized by sex.

	Male N = 64	Female N = 67	*p*	Total N = 113	Total Min.–Max.
Weight (kg)	45.33 (12.04)	45.87 (10.96)	0.52	45.61 (11.46)	25.9–87
Height (cm)	153.09 (9.16)	152.78 (7.12)	0.928	152.93 (8.15)	134.2–173
BMI (score)	19.08 (3.77)	19.53 (4)	0.493	19.31 (3.89)	13–33
PHV (years)	13.99 (0.69)	12.29 (0.7)	<0.001 *	13.12 (1.1)	11–15.7
Age (years)	12.49 (1.09)	12.66 (1.08)	0.323	12.57 (1.08)	10.8–15.9
Agility (s)	7.44 (0.69)	8.15 (0.82)	<0.001 *	7.8 (0.84)	5.1–10.9
TTD ^1^ (s)	19.43 (2.77)	19.54 (2.77)	0.814	19.48 (2.76)	13.9–28.1
LLEST (cm)	134.75 (30.7)	118.24 (24.35)	0.001 *	126.31 (28.75)	45.6–280
ULEST (cm)	260.82 (64.52)	233.98 (65.77)	0.014	247.09 (66.29)	70.5–450.3
FitC ^1^ (score)	230.16 (73.44)	174.18 (73)	0.001 *	201.53 (78.16)	20.61–387.4
Corsi (score)	37.13 (15.71)	32.2 (14.82)	0.038	34.61 (15.4)	2–104
Digit span (score)	8.09 (1.55)	8.39 (1.79)	0.131	8.24 (1.68)	5–13
HFT (cost)	10.41 (2.56)	10.85 (3.03)	0.15	10.64 (2.81)	4.61–26.2
SPT-R (cost)	58.8 (7.38)	61.38 (6.4)	0.041	60.12 (6.99)	37–72.13
Port. (score)	5.33 (1.56)	6.08 (1.4)	0.015	5.71 (1.52)	1.14–9.38
Math ^1^ (score)	6.68 (1.37)	6.35 (1.52)	0.193	6.51 (1.45)	3–9.9
Grades. (score)	6.49 (0.97)	6.57 (1.07)	0.989	6.53 (1.02)	4.2–9.2
AcadC (score)	7.44 (0.67)	7.67 (0.71)	0.211	7.56 (0.7)	5.6–9.46

Notes: Values expressed as mean and standard deviation. Lower values of s (seconds) and c (cost) correspond to better results; * significance level *p* of the Mann–Whitney test (except ^1^ = *t*-test), after Bonferroni correction = *p* < 0.003; BMI (body mass index); PHV (peak high velocity); TTD (disc touch test); LLEST (lower limb explosive strength test); ULEST (upper limb explosive strength test); FitC (physical fitness composite score); Corsi (total score of the Corsi’s block task); digit span (sum score up to two errors of the direct and indirect order digit span task); HFT (cost of accurate responses of the hearts and flowers—executive functions task); SPT-R (school performance test of reading); Port. (Portuguese); math (mathematics); grades (overall school grades); AcadC (academic performance composite score).

**Table 2 behavsci-14-01022-t002:** Correlations between physical and cognitive tests and academic performance.

	BMI	Age	Agility (s)	TTD ¹ (s)	LLEST	ULEST	FitC ¹
Corsi	0.095 (0.281)	0.004 (0.963)	−0.200 (0.022)	−0.184 (0.036)	0.111 (0.208)	0.192 (0.028)	0.250 (0.004)
Digit	−0.01 (0.87)	−0.046 (0.601)	−0.139 (0.114)	−0.115 (0.193)	−0.08 (0.368)	−0.017 (0.849)	0.06 (0.496)
HFT (c)	−0.096 (0.273)	0.084 (0.341)	0.420 * (<0.001)	0.336 * (<0.001)	−0.229 (0.009)	−0.245 (0.005)	−0.457 * (<0.001)
SPT-R	−0.085 (0.332)	−0.047 (0.598)	0.019 (0.832)	−0.126 (0.15)	0.018 (0.835)	−0.145 (0.098)	0.011 (0.897)
Portuguese	−0.025 (0.776)	−0.183 (0.036)	0.203 (0.020)	0.213 (0.015)	−0.254 (0.003)	−0.301 * (<0.001)	−0.365 * (<0.001)
Mathematics ^1^	−0.183 (0.036)	−0.294 * (0.001)	−0.210 (0.016)	−0.190 (0.03)	0.175 (0.045)	0.044 (0.616)	0.206 (0.018)
Grades	−0.156 (0.076)	−0.234 (0.007)	−0.062 (0.484)	−0.031 (0.724)	0.03 (0.73)	−0.093 (0.29)	0.008 (0.925)
AcadC	−0.15 (0.087)	−0.169 (0.054)	−0.094 (0.283)	−0.178 (0.042)	0.04 (0.65)	−0.13 (0.138)	0.077 (0.379)

Notes: Measures are expressed as mean and standard deviation. Smaller values for s (seconds) and c (cost) correspond to better results. The remaining variables are scores, with higher scores corresponding to better results. ^1^—Pearson correlations. The remaining correlations are Spearman’s; * significance level of *p* after Bonferroni correction ≤ 0.003; BMI (body mass index); PHV (peak high velocity); TTD (touch disc test); LLEST (lower limb explosive strength test—result in centimeters); ULEST (upper limb explosive strength test—result in centimeters); FitC (fitness composite score); Corsi (total score of the Corsi´s block task); digit (sum score up to two errors of the forward and backward digit span); HFT (cost of accurate responses of the hearts and flowers executive functions task); SPT-R (reading school performance test); grades (overall school grades); AcadC (composite score of academic performance).

**Table 3 behavsci-14-01022-t003:** Analysis of correlations controlled for age and sex, between physical, cognitive, and academic performance variables.

	BMI		Agility (s)		TTD (s)		LLEST		ULEST		FitC	
	Imp	Orig	Imp	Orig	Imp	Orig	Imp	Orig	Imp	Orig	Imp	Orig
Corsi	0.089 (0.313)	0.097 (0.366)	−0.074 (0.407)	0.135 (0.372)	−0.122 (0.17)	−0.129 (0.227)	0.065 (0.463)	0.042 (0.713)	0.096 (0.278)	0.089 (0.453)	0.131 (0.138)	−0.081 (0.616)
Digit	−0.021 (0.815)	−0.026 (0.772)	−0.179 (0.043)	0.019 (0.889)	−0.115 (0.195)	−0.106 (0.245)	−0.026 (0.774)	−0.062 (0.518)	−0.006 (0.947)	−0.033 (0.739)	0.15 (0.092)	0.045 (0.755)
HFT (c)	−0.115 (0.195)	−0.134 (0.145)	0.494 * (<0.001)	0.384 (0.004)	0.373 * (<0.001)	0.348 * (<0.001)	−0.253 (0.004)	−0.186 (0.055)	−0.238 (0.007)	−0.169 (0.092)	−0.525 * (<0.001)	−0.411 (0.004)
SPT-R	−0.095 (0.283)	−0.077 (0.394)	−0.083 (0.348)	0.004 (0.974)	−0.173 (0.05)	−0.155 (0.087)	0.07 (0.427)	0.024 (0.802)	−0.085 (0.341)	−0.097 (0.325)	0.085 (0.34)	0.089 (0.535)
Port.	0.03 (0.738)	−0.037 (0.768)	0.034 (0.704)	−0.034 (0.842)	0.229 (0.009)	−0.054 (0.666)	−0.059 (0.504)	0.138 (0.323)	−0.171 (0.053)	−0.142 (0.347)	−0.252 (0.004)	−0.213 (0.267)
Math	−0.133 (0.132)	−0.216 (0.082)	−0.174 (0.049)	−0.096 (0.578)	−0.2 (0.023)	−0.226 (0.068)	0.14 (0.115)	0.254 (0.066)	0.07 (0.432)	−0.057 (0.708)	0.246 (0.005)	0.13 (0.501)
Grades	−0.087 (0.328)	−0.129 (0.261)	−0.08 (0.369)	−0.008 (0.961)	−0.032 (0.722)	−0.104 (0.364)	0.11 (0.217)	0.103 (0.416)	−0.051 (0.564)	−0.212 (0.11)	0.061 (0.489)	−0.073 (0.697)
AcadC	−0.134 (0.13)	−0.2 (0.088)	−0.12 (0.175)	0.078 (0.656)	−0.15 (0.089)	−0.166 (0.158)	0.133 (0.134)	0.137 (0.292)	−0.1 (0.26)	−0.184 (0.183)	0.108 (0.225)	−0.081 (0.684)

Notes: Measures are expressed as mean and standard deviation. Smaller values for s (seconds) and c (cost) correspond to better results. The remaining variables are scores, with higher scores corresponding to better results. BMI (body mass index); PHV (peak high velocity); TTD (touch disc test); LLEST (lower limb explosive strength test); ULEST (upper limb explosive strength test); FitC (fitness composite score); Corsi (total score of the Corsi´s block task); digit (sum score up to two errors of the forward and backward digit span); HFT (cost of accurate responses of the hearts and flowers executive functions task); SPT-R (reading school performance test); Port. (Portuguese); grades (overall school grades); AcadC (composite score of academic performance); * significance level after Bonferroni correction *p* ≤ 0.003.

## Data Availability

The data presented in this study are available on request from the corresponding author due to restrictions imposed by the ethics committee.

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
