# Peer review of "Physical Fitness, Executive Functions, and Academic Performance in Children and Youth: A Cross-Sectional Study"

_behavsci, 2024, doi:10.3390/bs14111022_

Round 1
Reviewer 1 Report
Comments and Suggestions for Authors
Dear,
I would like to congratulate the authors on their motivation and interest in this work. The introduction of this study clearly outlines the problem in the researched area. The theoretical foundations of the investigated issue are presented, emphasizing the importance of physical activity and the negative implications of physical inactivity in children. Previous research from the same field is also mentioned in the introduction.
The aim of the study was to determine the relationship between physical fitness, executive functions, and academic achievement in children and adolescents. A total of 131 participants were included in the study, of which 49% were female and 51% male. A regression analysis was used to examine the relationship between executive functions, physical fitness, and academic achievement. All measurement procedures are described in detail. The results are presented in tables, with the most significant findings thoroughly analyzed. In the discussion, the most important results are compared with previous studies. The main advantages of the research, as well as the identified shortcomings, are also highlighted. The conclusion is appropriately structured and demonstrates high quality. At the end of the paper, practical implications are provided, which will be useful to future researchers who might design similar experimental studies aimed at promoting sports-related fitness skills to explore potential effects on children’s cognitive traits.
The manuscript is well-structured and relevant to the field of research. The literature used is mostly recent, except in the introduction, where the theoretical foundations of the research problem are discussed. This paper can certainly serve as a scientific basis for developing new research strategies aimed at encouraging educational policies to design programs that will impact the development of physical fitness and academic achievement in children.
Suggestions for changes:
- Specify the time period during which the research was conducted (year and month).
- Who were the evaluators? Were they trained beforehand, and what were their qualifications (school teachers or specially trained evaluators)?
Finally, I believe this article is acceptable for publication in the journal, and my suggestion is to accept the manuscript with minor corrections
Author Response
Comment 1: Specify the time period during which the research was conducted (year and month).
Response 1: Thank you for your careful review and for pointing this out. We have included the period information on page 3, lines 97 and 98 - "The assessments were conducted from February to the first week of March 2020."
Comment 2: Who were the evaluators? Were they trained beforehand, and what were their qualifications (school teachers or specially trained evaluators)?
Response 2: Thanks for your suggestion. We have included evaluator information on page 4, lines 173 to 175 - "The team of assessors was made up of psychologists and physical education teachers, who were properly trained in the instruments of this research."
Reviewer 2 Report
Comments and Suggestions for Authors
Please find the attachment.
